# Financing Cocaine Use in a Homeless Population

**DOI:** 10.3390/bs7040074

**Published:** 2017-10-25

**Authors:** Carol S. North, David E. Pollio

**Affiliations:** 1The Altshuler Center for Education & Research at Metrocare Services and the Department of Psychiatry, The University of Texas Southwestern Medical Center, 6363 Forest Park Rd., Suite BL6.226, Dallas, TX 75390-8828, USA; 2Department of Social Work, College of Arts and Sciences, University of Alabama Birmingham, Heritage Hall Building 322, 1720 2nd Avenue South, Birmingham, AL 35294-1152, USA; davpoll@uab.edu

**Keywords:** homelessness, substance use, cocaine, financing, income, public entitlements, longitudinal, diagnostic assessment, urine drug testing, panhandling

## Abstract

**Background:** Cocaine use is highly prevalent among homeless populations, yet little is known about how it is financed. This study examined associations of income sources with cocaine use and financing of drugs in a longitudinal evaluation of a homeless sample. **Methods:** A homeless sample was recruited systematically in St. Louis in 1999–2001 and longitudinally assessed annually over two years using the Diagnostic Interview Schedule and the Homeless Supplement, with urine drug testing. **Results:** More than half (55%) of participants with complete follow-up data (*N* = 255/400) had current year cocaine use. Current users spent nearly $400 (half their income) in the last month on drugs at baseline. Benefits, welfare, and disability were negatively associated and employment and income from family/friends, panhandling, and other illegal activities were positively associated with cocaine use and monetary expenditures for cocaine. **Conclusions**: Findings suggest that illegal and informal income-generating activities are primary sources for immediate gratification with cocaine use and public entitlements do not appear to be primary funding sources used by homeless populations. Policy linking drug testing to benefits is likely to have little utility, and public expenditures on measures to unlink drug use and income might be more effectively used to fund employment and treatment programs.

## 1. Introduction

Members of homeless populations by definition lack sufficient income to maintain secure housing. However, substance use and abuse are highly prevalent in homeless populations [1,2,3,4], implying that sources of funding must be available for obtaining substances of abuse. In particular, homeless populations have been demonstrated to have a very high prevalence of cocaine use [5,6,7,8]. Several studies have examined various sources of income in homeless populations and reported income to be associated with substance use. A homeless study conducted in Alameda County, California found that informal sources of income such as income from panhandling and other illegal activities to be the most frequent sources of current income, especially among those with current year substance use disorders [9]. Multivariate analysis of data from homeless studies conducted in Pittsburgh and Philadelphia found that alcohol and drug use disorders were associated with income from odd jobs and illegal activities [10]. Studies of street-involved youth have found that risky (illegal or quasi-legal) income-generating activities (e.g., sex work, salvaging/recycling, solicitation of donations for small services, panhandling, drug dealing, theft, and other criminal activities to generate income) are highly prevalent and are further associated with drug use [11,12,13].

These findings are consistent with research on the relationship of income and drug use in more general populations. Individuals addicted to narcotics are well known to derive the bulk of their income from criminal activities [14]. A multisite study of female crack cocaine users found that 70% of the women exchanged sex for drugs [15]. A multisite study of social security income recipients revealed that urine samples were 28% more often positive for cocaine in the first ten days of the month compared to the rest of the month, suggesting a temporal relationship between cocaine use and availability of income [16].

The findings from this limited body of research suggest the need for further exploration of the relationship between drug use and its financing specifically in homeless populations whose financial resources are ostensibly limited. Additionally, there is need to examine how cocaine in particular is financed in homeless populations, because of the remarkably high prevalence of cocaine use demonstrated in this population [7,8], in the context of the expense of this drug. Therefore, this study examined associations of income sources with cocaine use and monetary expenditures for drugs in a longitudinal evaluation of a systematically recruited sample of homeless people assessed annually over two years.

## 2. Methods

This prospective longitudinal study of literal homelessness systematically recruited a sample of 400 individuals from shelter and street locations to examine psychosocial correlates of the course of homelessness and service utilization over time. The cohort for this study includes 255 members of the original sample who were successfully re-assessed annually over two years.

### 2.1. Ethics

The project was approved in advance by the Washington University School of Medicine Institutional Review Board. All participants provided written informed consent before entering the study.

### 2.2. Sample

Participants for the study were systematically selected between October, 1999 and May, 2001 from shelter (80%) and street (20%) settings using randomly generated computerized schedules proportional to current shelter rosters or bed counts at each facility and across systematically traveled computer-randomized street routes [17,18]. To achieve the original sample of 400, 435 eligible homeless individuals were invited to participate in the study, but 35 did not enroll in the study, yielding 92% participation at baseline. More details of the sampling procedures and instruments of measure used in this study and the definition of homelessness applied are described in earlier publications by this research team [7,17,18].

Of the baseline sample of 400 participants, 288 (72%) were successfully tracked over two years and 257 (69%) were reassessed at both one and two years. There were no significant differences in terms of baseline demographic characteristics (gender, age, ethnic background, education, or marital status), chronicity of homelessness, income, legal problems, drug use, lifetime and current drug/alcohol disorder, and any other psychiatric disorder between those tracked and not tracked through the entire two years of study follow-up or between those with all three study assessments competed versus those missing one or more assessments. This report describes findings among the 255 individuals with complete data on all major variables at all three annual assessments, representing 59% of the original 435 individuals originally eligible for study participation.

### 2.3. Measures

At all three assessments (baseline, 12-month, and 24-month interviews), study participants were administered a structured interview using the Diagnostic Interview Schedule/Homeless Supplement [7]. The interview inquired about a list of sources and amounts of income from various sources in the last 30 days, amounts spent on drugs in the last 30 days, and types of substances used within the last year. Upon completion of each interview, participants provided a urine sample, which was qualitatively tested for the presence or absence of cocaine using the Roche OnTRAK test kit. This urine drug testing system has been documented to have a 4/1000 error rate and a reliable device for rapid onsite testing of urine specimens [19] and adequate performance in human service settings relative to drug screening assays of commercial laboratories [20].

At each assessment, participants were asked to indicate how much money they had earned in the last 30 days specifically from the following sources: employment earnings, money provided by family or friends, employment benefits (pension, benefits, and social security income), “welfare” (public assistance including Temporary Assistance for Needy Families, General Assistance, food stamps), social security disability, panhandling, and other illegal activities in the last 30 days. Dichotomous variables were constructed to represent earnings (yes/no) from each of these sources in the last 30 days. Combined variables were constructed to reflect any income (dichotomous data) from each of these sources in any of three 30-day time periods as well as the sum of amount of income (numerical variables) received from each of these sources across the three 30-day time periods.

Because this study found that cocaine was the most used and abused drug by this sample—exceeding cannabis and all other drugs [7]—this study’s investigation of the financing of drug use focused on cocaine. Comparison of the combined urine cocaine test results reflecting a positive test on any of the three test points (dichotomous variables) with the combined data reflecting any source of income received across the three assessments (dichotomous variables) and with the total amounts of 30-day income reported at all three interviews (numerical variables) permitted examination of the association of having any positive urine cocaine test with earning any income and amount of income earned in the three 30-day time periods over the course of the three study years. Because these variables did not indicate a specific intersection of urine cocaine test result and income at the same assessment point, they reflect general tendencies within individuals to have positive urine cocaine tests and to receive income from various sources across three years of assessments rather than momentary associations.

Additional variables were created to capture temporal associations of positive urine tests for cocaine and income received from each source in the same month. To accomplish this, at each of the three assessments, urine test results for cocaine (positive/negative) were matched with the presence or absence of reported income for each type of income source, yielding a total of 765 comparisons (3 for each of 255 individuals). At each of the three assessments, urine test results for cocaine were also matched with amount of income received from each source of income, again yielding a total of 765 comparisons. This matching of the urine cocaine test variables with income variables at each assessment point ensured that urine cocaine results were temporally matched with 30-day income variables at each time point. Because these variables indicate the intersection of urine cocaine test results with income received from various sources at the same time point, they reflect the tendency for detection of cocaine use and income to be associated at a single point in time. Similar variables and procedures were created to represent money spent on drugs in the last 30 days at each assessment point and to match them with specific sources of income received in the same time frame.

### 2.4. Data Analysis

Data analysis was conducted using SAS 9.4 (SAS Institute, Cary, NC, USA). To address missing data, the missing information was cross-referenced with self-report of service utilization, followed by a multiple-imputation process as described in detail in a previous publication [21]. Dichotomous variables were compared using χ^2^ tests, substituting two-tailed Fisher’s exact tests in instances of expected cell sizes <5. To address potential non-normality of data, nonparametric Wilcoxon rank sum tests (PROC NPAR1WAY in SAS) were used to compare dichotomous variables with numerical variables. Numerical variables were compared using Pearson correlations. Multivariate analysis does not permit nonparametric methods, and thus logistic regression models (PROC LOGISTIC in SAS) were used to further examine significant associations between income and drug use identified in nonparametric models by controlling for sex and age (variables showing significant bivariate associations with urine tests for cocaine and/or income variables). Level of statistical significance for all comparisons was set as α ≤ 0.05.

## 3. Results

At baseline, the study sample (*N* = 255) was 73% male, relatively young (median = 43 years of age), predominantly African–American (77%), high school educated (median years of education = 12), unemployed (70%), and never-married (54%). The mean (SD) years of lifetime homelessness was 4.4 (5.4) and the median was 3. The mean (SD) number of lifetime homeless episodes was 2.6 (3.4). Overall, 93% had received some form of income in the last 30 days. The mean (SD) income in the last 30 days from all sources for the entire sample was $590 ($660). The main sources of income reported at baseline were welfare for 46%, employment for 41%, and disability income for 20%.

At baseline, based on combined data from urine drug testing and self-reported current year cocaine use, 20% had cocaine detected in their urine and 44% had evidence of cocaine use. More than one-third (38%) of the sample had cocaine detected in their urine on at least one of the three annual urine tests, and more than half (55%) of the sample had any positive evidence of current year cocaine use as determined by any positive urine test result or self report of current year cocaine use in any of the three study years. At baseline, 28% of the sample and 59% of participants with current year cocaine use reported having spent any money on drugs in the last 30 days, and most (87%) of those with drug expenditures reported current year cocaine use. The mean (SD) amount of money spent on drugs in the last 30 days was $383 ($574) among those with current-year cocaine use; the median amount was $130, the maximum was $3300, and 7 individuals spent >$1000 for drugs in the last month. Overall, 30-day drug expenditures at baseline represented 17% of the total 30-day income for the entire sample and 50% for those with current year cocaine use.

At baseline, men were more likely than women to have a positive urine drug test for cocaine (44% vs. 22%; χ^2^ = 10.24, *df* = 1, *p* = 0.001) and to be receiving income from earnings (49% vs. 18%; χ^2^ = 18.66, *df* = 1, *p* < 0.001), but less likely to be receiving income from welfare (62% vs. 81%; χ^2^ = 8.05, *df* = 1, *p* = 0.005). Receiving income was associated with younger age for money received from family or friends (mean = 39.4, SD = 11.6 for those receiving vs. mean = 42.5, SD = 9.2 for those not receiving income; Wilcoxon S = 9212, z = −1.94, *p* = 0.052) and with older age for receiving income from employment benefits (mean = 47.7, SD = 12.8 for those receiving vs. mean = 41.1, SD = 9.8 for those not receiving income; Wilcoxon S = 2543, z = 2.47, *p* = 0.014) and disability (mean = 46.0, SD = 10.6 vs. mean = 40.4, SD = 9.7; Wilcoxon S = 7612, z = 3.29, *p* = 0.001). Race was not significantly associated with any source of income. Neither age nor race was associated with a positive urine drug test for cocaine.

Income sources were compared with positive urine drug test results. Table 1 and Figure 1 present the proportions of those with and without positive urine cocaine tests at any of the three assessment points who reported types of any 30-day income received *at any time in the study*. A positive urine cocaine test was associated only with 30-day panhandling and receiving other illegal income in these comparisons. A series of separate multivariate logistic regression models (one for each income source as a dependent variable) including sex and age as independent covariates also found positive urine cocaine tests to be significantly associated with 30-day panhandling (β = 0.80, standard error [SE] = 0.36, Wald χ^2^ = 4.92, *p* = 0.027) and other illegal income (β = 1.09, SE = 0.39, Wald χ^2^ = 7.62, *p* = 0.006) at any time in the study.

Table 2 and Figure 2 present the proportions of those with any positive urine cocaine test who reported types of 30-day income received *in the same month*. A positive urine cocaine test was negatively associated with any 30-day disability income received and positively associated with panhandling and any 30-day illegal income received in these comparisons. A series of separate multivariate logistic regression models (one for each income source as a dependent variable) including sex and age as independent covariates also found positive urine cocaine tests to be significantly associated with any 30-day illegal income (β = 0.78, SE = 0.35, Wald χ^2^ = 5.05, *p* = 0.025), but the association with any 30-day disability income fell short of significance (*p* = 0.082) in the same month.

Table 3 and Figure 3 present the combined amounts of reported 30-day income from various sources at any of the three assessment points among those with and without a positive urine cocaine test *at any time in the study*. A positive urine cocaine test was marginally associated with lower amounts of 30-day income received from disability and significantly associated with higher amounts of 30-day income from panhandling and from illegal activities at any time in these comparisons. Separate multivariate logistic regression models (one for each income source as a dependent variable) including sex and age as independent covariates also found positive urine cocaine tests to be negatively associated with amount of 30-day income from disability (β = 0.00, SE = 0.00, Wald χ^2^ = 3.97, *p* = 0.046), but not associated with amount of 30-day panhandling or other illegal income at any time in the study; positive urine cocaine tests were also associated with amount of income from welfare (β = 0.00, SE = 0.00, Wald χ^2^ = 4.19, *p* = 0.041).

Table 4 and Figure 4 present the amounts of reported 30-day income from various sources among those with and without a positive urine cocaine test *in the same month*. A positive urine cocaine test was associated with lower amounts of 30-day income from disability and with higher amounts of 30-day income from illegal activities in these comparisons. A series of separate multivariate logistic regression models (one for each income source as a dependent variable) including sex and age as independent covariates also found positive urine cocaine tests to be negatively associated with amounts of 30-day income from disability (β = 0.00, SE = 0.00, Wald χ^2^ = 4.29, *p* = 0.038) but not associated with amounts of 30-day income from illegal activities in the same month.

Income sources were also compared with 30-day monetary expenditures for drugs. Table 5 and Figure 5 present the proportions of those reporting and not reporting 30-day drug expenditures at any of the three assessment points who reported types of 30-day income *at any time in the study*. Participants with 30-day monetary expenditures for drugs on any of the three assessments were less likely than others to report any 30-day disability income and more likely to report any 30-day panhandling income and any 30-day other illegal income on any of the three assessments. A series of separate multiple logistic regression models (one for each income source as a dependent variable) including sex and age as independent covariates also found 30-day drug expenditures to be associated with any income from panhandling (β = 1.18, SE = 0.41, Wald χ^2^ = 8.31, *p* = 0.004) and illegal behaviors (β = 2.74, SE = 0.64, Wald χ^2^ = 18.11, *p* < 0.001), but the association with any disability income fell short of significance (*p* = 0.061), at any time in the study.

Table 6 and Figure 6 present the proportions of those reporting and not reporting 30-day drug expenditures who reported types of 30-day income *in the same month*. Participants with 30-day monetary expenditures for drugs were more likely to have received any 30-day income from employment, less likely to have received any 30-day income from employment benefits and from disability benefits, and more likely to have received any 30-day income from panhandling and other illegal income in the same month. A series of separate multiple logistic regression models (one for each income source as a dependent variable) including sex and age as independent covariates also found any 30-day drug expenditures to be associated with any income received from panhandling (β = 1.13, SE = 0.29, Wald χ^2^ = 14.85, *p* < 0.001) and other illegal behaviors (β = 2.04, SE = 0.35, Wald χ^2^ = 33.61, *p* < 0.001) in the same month, but the association with any income from employment fell short of significance (*p* = 0.059) and associations with any income from employment benefits and disability benefits were not significant.

Table 7 and Figure 7 present the combined amounts of reported 30-day income from various sources at any of the three assessment points among those with and without a reported 30-day drug expenditure *at any time in the study*. Reported 30-day drug expenditure was associated with higher amounts of 30-day income from employment, panhandling, and other illegal activities, and lower amounts of 30-day income from disability benefits on any of the three assessments. A series of separate multiple logistic regression models (one for each income source as a dependent variable) including sex and age as independent covariates also found any 30-day drug expenditures to be negatively associated with amount of disability income (β = 0.00, SE = 0.00, Wald χ^2^ = 6.43, *p* = 0.011) on any of the three assessments, but not with amounts of 30-day income from employment (*p* = 0.411) or illegal activities (*p* = 0.076).

Table 8 and Figure 8 present the combined amounts of reported 30-day income from various sources among those with and without reported 30-day drug expenditure *in the same month*. Reported 30-day drug expenditure was positively associated with amounts of income from employment, panhandling, and other illegal activities, and negatively associated with amounts of income from employment benefits, welfare, and disability benefits in the same month. A series of separate multiple logistic regression models (one for each income source as a dependent variable) including sex and age as independent covariates also found any 30-day drug expenditures to be associated with amount of income received from panhandling (β = 0.00, SE = 0.00, Wald χ^2^ = 4.09, *p* = 0.043) and other illegal activities (β = 0.00, SE = 0.00, Wald χ^2^ = 11.36, *p* < 0.001), but the association with amount of disability income fell short of significance (*p* = 0.056) and amount of income received from employment and from employment benefits were not associated.

The sum of 30-day monetary expenditures for drugs reported across the three assessments was associated with the combined amounts of reported 30-day income *across all three assessments* obtained from family/friends (*r* = 0.18, *p* = 0.009), panhandling (*r* = 0.56, *p* < 0.001), and other illegal income (*r* = 0.43, *p* < 0.001). In a series of separate multiple logistic regression models (one for each income source as a dependent variable) including sex and age as independent covariates also found amount of 30-day drug expenditures to be associated with any income received from panhandling (β = 6.14, SE = 0.63, *df* = 1, *t* = 9.70, *p* < 0.001) and other illegal behaviors (β = 2.52, SE = 0.35, *df* = 1, *t* = 7.29, *p* < 0.001) on any of the three assessments, but the association with any income from family/friends was not significant. The amount of 30-day monetary expenditure for drugs at any of the three assessments was associated with the amount of reported 30-day income *in the same month* obtained from family/friends (*r* = 0.15, *p* < 0.001), panhandling (*r* = 0.38, *p* < 0.001), and other illegal income (*r* = 0.26, *p* < 0.001). In a series of separate multiple logistic regression models (one for each income source as a dependent variable) including sex and age as independent covariates also found amount of 30-day drug expenditures to be associated with amount of income received from panhandling (β = 3.77, SE = 0.25, *df* = 1, *t* = 15.26, *p* < 0.001) and other illegal behaviors (β = 0.40, SE = 0.07, *df* = 1, *t* = 5.64, *p* < 0.001) in the same month, but not with amount of income from family/friends.

A specific focus on panhandling found that among individuals reporting 30-day income from panhandling at any of the three study assessments, *same-month* evidence of drug use was found in 83% and of cocaine use specifically in 68%. Further, 50% of those with 30-day panhandling at any study assessment also reported spending money for drugs in the same month that they panhandled, and their current month average panhandling income ($185) represented 28% of their current month drug expenditures ($668).

## 4. Discussion

More than one-third (38%) of this homeless sample had cocaine detected in their urine on at least one of the three annual urine tests, and more than half (55%) of the sample had any positive evidence of current year cocaine use based on combined data representing any positive urine test result or self report of current year cocaine use at any of the three study assessments. Current year cocaine users reported spending nearly $400 in the last month on drugs at baseline, representing half of their total month’s income.

The main overall findings of these analyses are that employment and income from family/friends were positively associated; benefits, welfare, and disability were negatively associated; and panhandling and other illegal activities were positively associated with cocaine use and monetary expenditures for cocaine in the various comparisons made. The analysis predicting cocaine use and income sources associated with it were consistent with each other: no significant associations of any income source with cocaine use or financing variable were significant in opposite directions. The findings from these analyses suggest that income from public sources (employment benefits, welfare, and disability) are unlikely to be used to finance cocaine use, as opposed to employment and income from informal and illegal sources which were found to support financing of cocaine use. These results are generally consistent with those of previous studies demonstrating associations of drug use in homeless populations with personal earnings, public entitlement income, monetary handouts, and illegal sources of income [4,9,10,11,12,13,14,22], and specifically studies also demonstrating significant associations with drug-related risky behaviors [23,24].

The difference in the direction of the significant association with cocaine use and income source with employment versus employment benefits likely reflects the type of work represented. Employment not accompanied by benefits is likely to represent work such as day labor, shadow work, etc. which are more ideally suited to rapid acquisition of funding when drug purchases are desired. Employment with benefits is more likely to be stable and require the ability to consistently show up for work not under the influence of substances, which would not be consistent with a cocaine habit. Borrowing from family or friends may have similarities to short-term employment in that it is a rapid source of money for purchasing drugs.

Concerns have been raised about the potential for diversion of taxpayer funding of welfare and disability benefits for homeless populations into resources for substance use [4]. The findings from this study suggest that this concern is unwarranted; in fact, the opposite appears to be true. These associations do not necessarily imply causality: it is possible that individuals not using cocaine are more likely to receive such benefits, and conversely, it is possible that individuals receiving such benefits are less likely to use them inappropriately to finance cocaine use. Either way, these findings indicate that public funding did not appear to promote cocaine use in this sample. These results are consistent with a recent study of military veterans experiencing homelessness that did not find drug use to be associated with public welfare support (including unemployment, welfare, and disability benefits) or with disability benefits [4].

Additional concerns have been raised that income from panhandling in homeless populations may be used to fund the drug habits that are so prevalent in this population. Over the course of the study, cocaine use and expenditures for drugs were observed in the majority of all months in which panhandling occurred, and panhandling income contributed to more than one-fourth of their drug expenditures in panhandling months. Panhandling was similar to other illegal activities in these analyses in regard to being clearly associated with substance use and its financing. Collectively, these findings confirm concerns that income from panhandling and other illegal income has significant risk of supporting drug habits and also counter popular concerns about diversion of public welfare and disability benefits into substance use.

This study’s findings can help inform decisions regarding placement of charitable and public contributions to help the homeless. For example, the findings that 83% of those who panhandled had current month drug (predominantly cocaine) use in this sample provides specific relevance for individual encounters with homeless individuals requesting monetary support in public settings by suggesting the likelihood of diversion of these contributions into drug use, especially cocaine use. Organizations that serve homeless populations, such as shelters, soup kitchens, and treatment programs, provide opportunities for charitable contributions to this population that circumvent the risks of supporting the drug use that appears to be inherent in panhandling donations to members of homeless populations.

The strengths of the current study include the systematic sampling of the homeless sample from shelters and street settings, its prospective longitudinal design with three annual assessments, lack of identifiable follow-up attrition bias, assessment of drug use not just in the context of substance use disorders, combining self-report and drug testing data on substance use, systematic assessment of income and amount of income received from various sources, matching drug use and expenditures with sources of income in the same month, and appropriate use of nonparametric data analysis. Limitations include the collection of data in one city that may not represent findings in other geographical locations and the timing of the data collection approximately a decade and a half ago that may not represent current associations of income and drug abuse in the homeless population. Patterns of substance use and type of substances used by homeless populations have been demonstrated to change considerably across previous decades [7]. Drug use and abuse are also known to vary considerably across different geographical locations and settings, relative to local drug demand, supply, associated harms of use, and harm reduction services [25,26].

Despite the age of the data, the findings represent new knowledge not previously reported. These data have established a parameter to ground further consideration and a model for conceptualization of further research that is needed into the financing of substance use. Future studies of the financing of substance use in homeless populations need to examine substance use patterns in newly collected datasets, in different settings and locations, and assessing additional commonly abused substances including alcohol, marijuana, new emerging drugs of abuse, and—especially in light of the recent increase in opiate addiction in the US—the opiate class of drugs.

The findings from this study represent associations between variables and do not provide a basis for conclusions regarding causation. Drug use was underdetected in this study, based on 31% of those with a positive urine drug test denying any drug use in the last month [8], suggesting that reported expenditures for drugs were also likely underestimated. Not all associations of positive urine cocaine tests with amount of income from panhandling and other illegal activities were present in multiple regression models controlling for sex and age, possibly reflecting the strength of sex and age in association with cocaine use and/or income obtained from these sources.

Financing cocaine use in homeless populations is not dissimilar to strategies used in general populations to fund cocaine use. Illegal and informal income-generating activities are primary sources for the immediate gratification of cocaine use. Public entitlements do not appear to be primary funding sources used by homeless populations, similar to the general population [14]. The data in this study do not appear to fulfill concerns that homeless people will use public welfare and disability income to fund their cocaine use; thus, policy linking drug testing to these benefits is likely to have little utility. However, public expenditures on measures to unlink drug use and income might be more effectively used to fund treatment programs that transition users of cocaine to stable employment (e.g., the type accompanied by benefits) or programs to help individuals with disabilities that preclude gainful employment.

## Figures and Tables

**Figure 1 behavsci-07-00074-f001:**
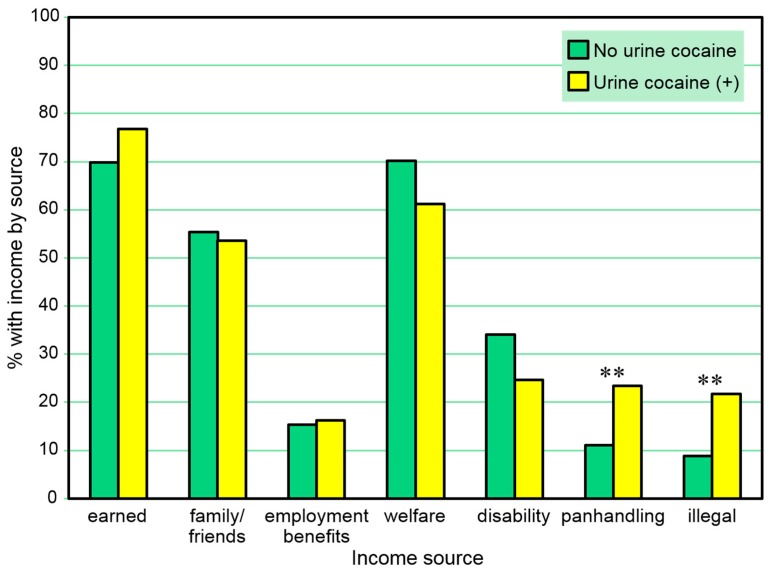
Proportions with 30-day income sources at any time point among those with and without a positive urine cocaine test at any time point. ** *p* ≤ 0.01.

**Figure 2 behavsci-07-00074-f002:**
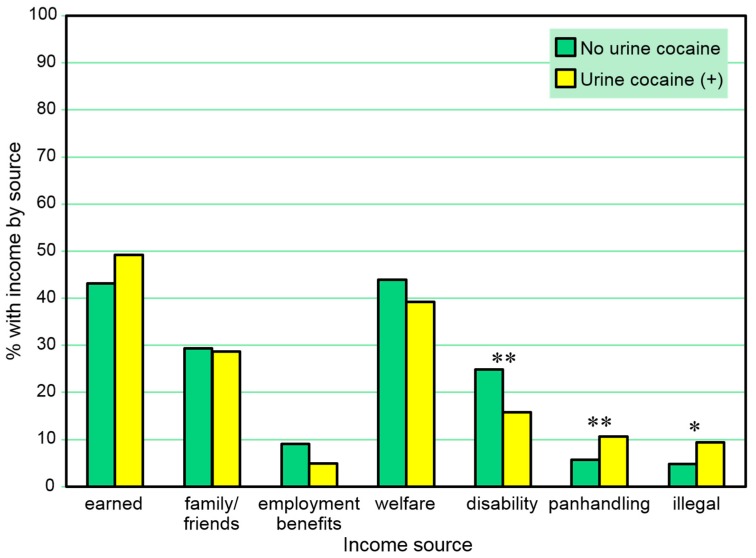
Proportions with 30-day income sources among those with and without a positive urine cocaine test in the same month. * *p* ≤ 0.05, ** *p* ≤ 0.01.

**Figure 3 behavsci-07-00074-f003:**
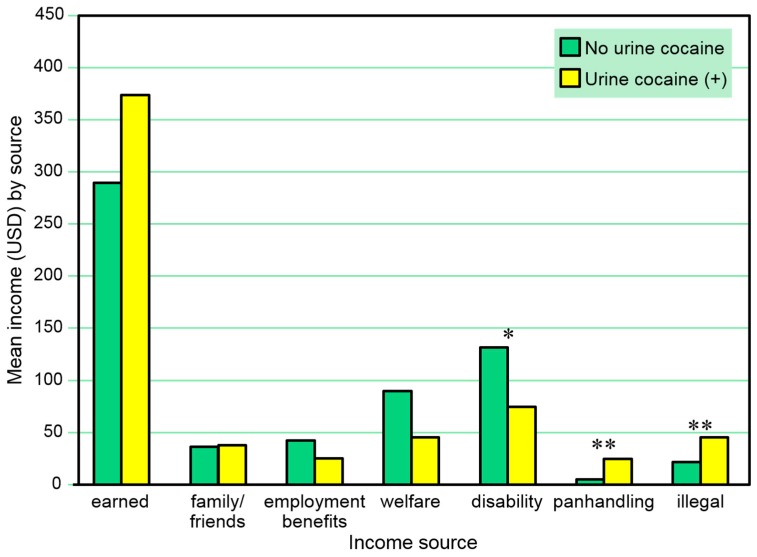
Total amounts earned from 30-day income sources across three interviews among those with and without a positive urine cocaine test at any time point. * *p* ≤ 0.05, ** *p* ≤ 0.01.

**Figure 4 behavsci-07-00074-f004:**
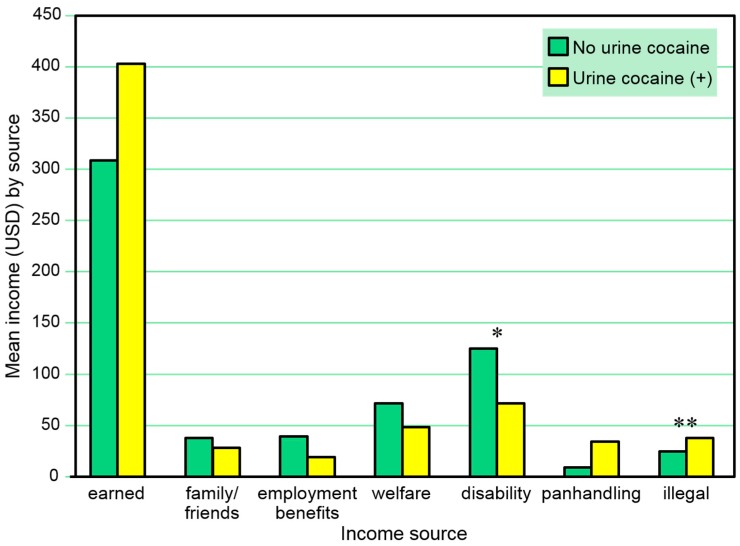
Amounts earned from various 30-day income sources among those with and without a positive urine cocaine test in the same month. * *p* ≤ 0.05, ** *p* ≤ 0.01.

**Figure 5 behavsci-07-00074-f005:**
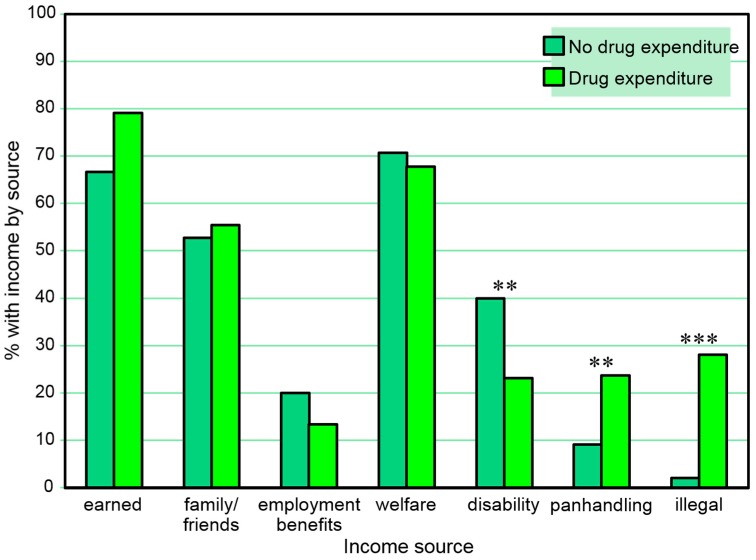
Proportions with any 30-day income sources among those with and without any 30-day drug expenditures across three interviews. ** *p* ≤ 0.01, *** *p* ≤ 0.001.

**Figure 6 behavsci-07-00074-f006:**
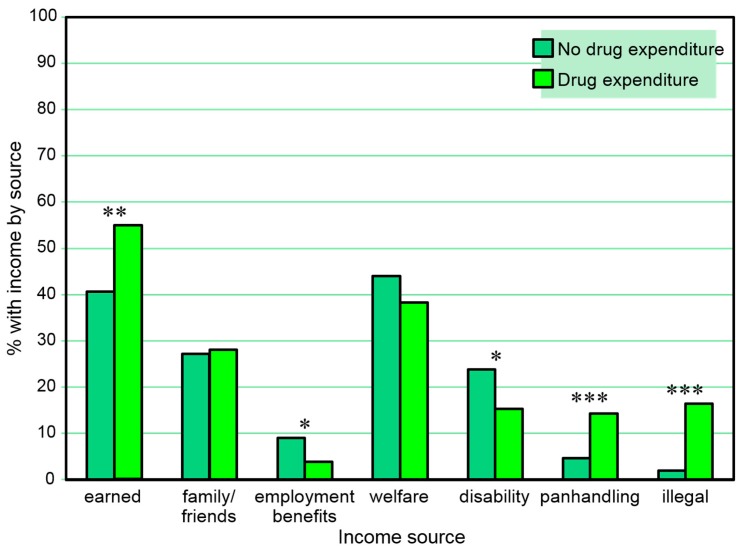
Proportions with 30-day income sources among those with and without 30-day drug expenditures in the same month. * *p* ≤ 0.05, ** *p* ≤ 0.01, *** *p* ≤ 0.001.

**Figure 7 behavsci-07-00074-f007:**
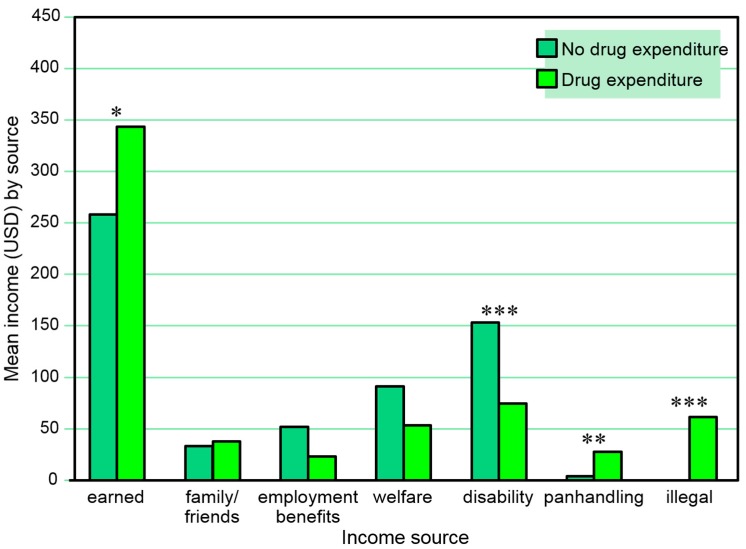
Total amounts earned from various 30-day income sources among those with and without 30-day drug expenditures across three interviews. * *p* ≤ 0.05, ** *p* ≤ 0.01, *** *p* ≤ 0.001.

**Figure 8 behavsci-07-00074-f008:**
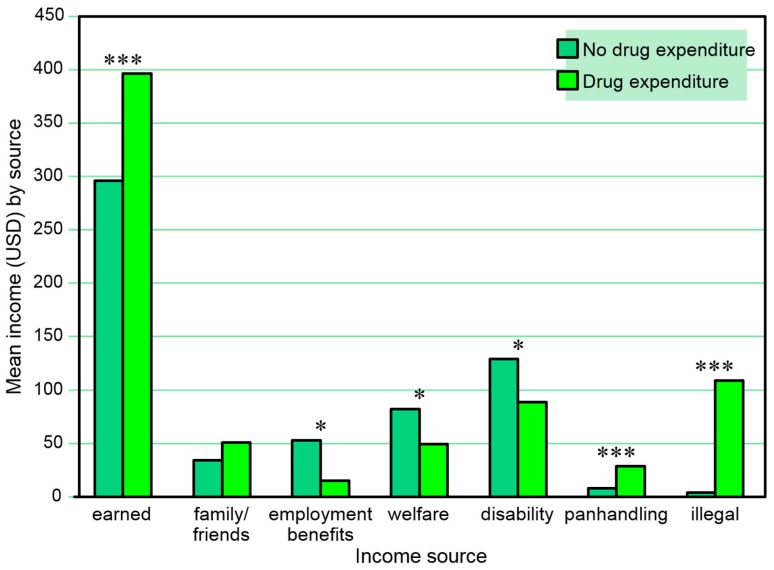
Amounts earned from various 30-day income sources among those with and without 30-day drug expenditures in the same month. * *p* ≤ 0.05, *** *p* ≤ 0.001.

**Table 1 behavsci-07-00074-t001:** Proportions with 30-day income sources at any time point among those with and without a positive urine cocaine test at any time point.

Income Source	No Urine Cocaine % (*n*)	Urine Cocaine (+) % (*n*)	Significance
Earned	70 (108)	77 (75)	χ^2^ = 1.56, *df* = 1, *p* = 0.212
Family/friends	56 (85)	54 (52)	χ^2^ = 0.09, *df* = 1, *p* = 0.763
Employment benefits	16 (24)	17 (16)	χ^2^ = 0.04, *df* = 1, *p* = 0.855
Welfare	70 (109)	61 (59)	χ^2^ = 2.10, *df* = 1, *p* = 0.147
Disability	34 (53)	25 (24)	χ^2^ = 2.46, *df* = 1, *p* = 0.117
Panhandling	11 (17)	24 (23)	χ^2^ = 6.89, *df* = 1, *p* = 0.009
Illegal activities	9 (13)	22 (21)	χ^2^ = 9.05, *df* = 1, *p* = 0.003

**Table 2 behavsci-07-00074-t002:** Proportions with 30-day income sources among those with and without a positive urine cocaine test in the same month.

Income Source	No Urine Cocaine % (*n*)	Urine Cocaine (+) % (*n*)	Significance
Earned	42 (197)	49 (141)	χ^2^ = 2.82, *df* = 1, *p* = 0.093
Family/friends	29 (134)	27 (79)	χ^2^ = 0.20, *df* = 1, *p* = 0.653
Employment benefits	8 (39)	6 (18)	χ^2^ = 1.20, *df* = 1, *p* = 0.274
Welfare	44 (205)	39 (114)	χ^2^ = 1.61, *df* = 1, *p* = 0.205
Disability	25 (116)	16 (46)	χ^2^ = 8.75, *df* = 1, *p* = 0.003
Panhandling	6 (26)	11 (33)	χ^2^ = 8.36, *df* = 1, *p* = 0.004
Illegal activities	5 (21)	9 (25)	χ^2^ = 5.35, *df* = 1, *p* = 0.021

**Table 3 behavsci-07-00074-t003:** Total amounts earned from 30-day income sources across three interviews among those with and without a positive urine cocaine test at any time point.

Income Source	No Urine Cocaine Mean (SD)	Urine Cocaine (+) Mean (SD)	Significance
Earned	287 (428)	374 (497)	Wilcoxon S = 13339, z = 1.73, *p* = 0.085
Family/friends	38 (89)	41 (115)	Wilcoxon S = 12331, z = 0.07, *p* = 0.947
Employment benefits	44 (125)	26 (76)	Wilcoxon S = 12360, z = 0.02, *p* = 0.983
Welfare	88 (140)	44 (48)	Wilcoxon S = 11375, z = 1.78, *p* = 0.075
Disability	134 (216)	77 (165)	Wilcoxon S = 11471, z = 1.94, *p* = 0.053
Panhandling	6 (37)	28 (118)	Wilcoxon S = 13382, z = 2.81, *p* = 0.005
Illegal activities	23 (159)	44 (146)	Wilcoxon S = 13396, z = 3.05, *p* = 0.002

**Table 4 behavsci-07-00074-t004:** Amounts earned from various 30-day income sources among those with and without a positive urine cocaine test in the same month.

Income Source	No Urine Cocaine Mean (SD)	Urine Cocaine (+) Mean (SD)	Significance
Earned	312 (489)	405 (746)	Wilcoxon S = 51379, z = 1.74, *p* = 0.083
Family/friends	42 153)	32 106)	Wilcoxon S = 48173, z = 0.02, *p* = 0.990
Employment benefits	43 (161)	21 (96)	Wilcoxon S = 40712, z = 1.17, *p* = 0.243
Welfare	74 (138)	49 (66)	Wilcoxon S = 47526, z = 0.33, *p* = 0.739
Disability	124 (251)	72 (189)	Wilcoxon S = 44927, z = 2.12, *p* = 0.034
Panhandling	10 (84)	31 (151)	Wilcoxon S = 49817, z = 1.69, *p* = 0.091
Illegal activities	29 (272)	39 (202)	Wilcoxon S = 50435, z = 2.69, *p* = 0.007

**Table 5 behavsci-07-00074-t005:** Proportions with any 30-day income sources among those with and without any 30-day drug expenditures across three interviews.

Income Source	No Drug Expenditures % (*n*)	Drug Expenditures % (*n*)	Significance
Earned	67 (83)	79 (88)	χ^2^ = 3.64, *df* = 1, *p* = 0.056
Family/friends	53 (65)	56 (62)	χ^2^ = 0.21, *df* = 1, *p* = 0.644
Employment benefits	20 (25)	13 (14)	χ^2^ = 2.49, *df* = 1, *p* = 0.014
Welfare	71 (88)	68 (75)	χ^2^ = 0.32, *df* = 1, *p* = 0.572
Disability	40 (49)	23 (26)	χ^2^ = 7.22, *df* = 1, *p* = 0.007
Panhandling	9 (11)	24 (27)	χ^2^ = 9.98, *df* = 1, *p* = 0.002
Illegal activities	2 (3)	28 (31)	χ^2^ = 30.32, *df* = 1, *p* < 0.001

**Table 6 behavsci-07-00074-t006:** Proportions with 30-day income sources among those with and without 30-day drug expenditures in the same month.

Income Source	No Urine Cocaine % (*n*)	Urine Cocaine (+) % (*n*)	Significance
Earned	41 (221)	55 (100)	χ^2^ = 9.69, *df* = 1, *p* = 0.002
Family/friends	27 (147)	28 (52)	χ^2^ = 0.06, *df* = 1, *p* = 0.807
Employment benefits	9 (47)	4 (7)	χ^2^ = 4.82, *df* = 1, *p* = 0.029
Welfare	44 (237)	38 (69)	χ^2^ = 2.42, *df* = 1, *p* = 0.119
Disability	24 (127)	16 (29)	χ^2^ = 5.04, *df* = 1, *p* = 0.025
Panhandling	5 (28)	15 (28)	χ^2^ = 19.22, *df* = 1, *p* < 0.001
Illegal activities	2 (13)	18 (32)	χ^2^ = 52.49, *df* = 1, *p* < 0.001

**Table 7 behavsci-07-00074-t007:** Total amounts earned from various 30-day income sources among those with and without 30-day drug expenditures across three interviews.

Income Source	No Urine Cocaine Mean (SD)	Urine Cocaine (+) Mean (SD)	Significance
Earned	257 (386)	345 (460)	Wilcoxon S = 14632, z = 2.27, *p* = 0.023
Family/friends	36 (85)	42 (119)	Wilcoxon S = 13625, z = 0.36, *p* = 0.722
Employment benefits	54 (129)	24 (89)	Wilcoxon S = 12816, z = 1.85, *p* = 0.064
Welfare	92 (148)	54 (68)	Wilcoxon S = 12870, z = 1.11, *p* = 0.266
Disability	158 (228)	75 (162)	Wilcoxon S = 12111, z = 3.07, *p* = 0.001
Panhandling	3 (21)	28 (115)	Wilcoxon S = 14548, z = 3.27, *p* = 0.005
Illegal activities	0 (2)	60 (226)	Wilcoxon S = 15239, z = 5.58, *p* < 0.001

**Table 8 behavsci-07-00074-t008:** Amounts earned from various 30-day income sources among those with and without 30-day drug expenditures in the same month.

Income Source	No Urine Cocaine Mean (SD)	Urine Cocaine (+) Mean (SD)	Significance
Earned	291 (596)	395 (595)	Wilcoxon S = 73054, z = 0.3.34, *p* < 0.001
Family/friends	36 (116)	51 (204)	Wilcoxon S = 66489, z = 0.37, *p* = 0.714
Employment benefits	54 (164)	17 (93)	Wilcoxon S = 63336, z = 2.21, *p* = 0.027
Welfare	82 (150)	50 (93)	Wilcoxon S = 61073, z = 2.16, *p* = 0.031
Disability	129 (257)	78 (190)	Wilcoxon S = 61624, z = 2.33, *p* = 0.020
Panhandling	8 (78)	29 (119)	Wilcoxon S = 70781, z = 4.43, *p* < 0.001
Illegal activities	6 (55)	113 (496)	Wilcoxon S = 73120, z = 7.30, *p* < 0.001

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
