# Peer review of "Financing Cocaine Use in a Homeless Population"

_behavsci, 2017, doi:10.3390/bs7040074_

Round 1

Reviewer 1 Report

The authors have made considerable changes to the paper. They have been very responsive to the reviewer comments. The paper is very unique and extends the existing knowledge. I congratulate the authors and recommend publication of this paper in its current form.

Reviewer 2 Report

Major concerns:

1.         Table and figure represented the same information which is not allowed

2.         In the figures there are no error bars how to get significance?

Minor concerns:

1.         Line 34 What is the full name of “NIDA”?

2.         Line 69 Which country is the kit bought from and what is the full name of OnTRAK?

3.         Line 122 α≤.05 is not in accordance with other part of the paper which uses βor p and causes confusion. At the same time α=.05 can not be considered significant.

.05 should be 0.05. Same condition happened in other places.

4.         Line 127 What is the meaning of “SD”?

5.         Line 134 “44% had evidence of cocaine use” is not clear, what is the evidence?

6.         Line 151 What are the meanings of S, z ?

7.         Line 164 What are the meanings of β?

8.         Line 375 “age of the data” causes confusion.

9.         Line 381 “-“ causes confusion. Same condition happened in other parts of the paper.